# Similarities and differences in spatial and non-spatial cognitive maps

Charley M. Wu [1,2]*, Eric Schulz[3], Mona M. Garvert[4,5,6], Björn Meder[2,7,8], Nicolas W. Schuck[5,9]

**1** Department of Psychology, Harvard University, Cambridge, Massachusetts, United States of America, **2** Center for Adaptive Rationality, Max Planck Institute for Human Development, Berlin, Germany, **3** Max Planck Research Group Computational Principles of Intelligence, Max Planck Institute for Biological Cybernetics, Tübingen, Germany, **4** Department of Psychology, Max Planck Institute for Human Cognitive and Brain Sciences, Leipzig, Germany, **5** Max Planck Research Group NeuroCode, Max Planck Institute for Human Development, Berlin, Germany, **6** Wellcome Centre for Integrative Neuroimaging, University of Oxford, John Radcliffe Hospital, Oxford, United Kingdom, **7** Max Planck Research Group iSearch, Max Planck Institute for Human Development, Berlin, Germany, **8** Department of Psychology, University of Erfurt, Erfurt, Germany, **9** Max Planck UCL Centre for Computational Psychiatry and Ageing Research, Berlin, Germany

* charleywu@fas.harvard.edu

**Data Availability Statement:** All data and analysis code is available from https://github.com/charleywu/cognitivemaps.

**Funding:** ES is supported by the Max Planck Society and the Jacobs Foundation. The funders

## Abstract

Learning and generalization in spatial domains is often thought to rely on a "cognitive map", representing relationships between spatial locations. Recent research suggests that this same neural machinery is also recruited for reasoning about more abstract, conceptual forms of knowledge. Yet, to what extent do spatial and conceptual reasoning share common computational principles, and what are the implications for behavior? Using a within-subject design we studied how participants used spatial or conceptual distances to generalize and search for correlated rewards in successive multi-armed bandit tasks. Participant behavior indicated sensitivity to both spatial and conceptual distance, and was best captured using a Bayesian model of generalization that formalized distance-dependent generalization and uncertainty-guided exploration as a Gaussian Process regression with a radial basis function kernel. The same Gaussian Process model best captured human search decisions and judgments in both domains, and could simulate realistic learning curves, where we found equivalent levels of generalization in spatial and conceptual tasks. At the same time, we also find characteristic differences between domains. Relative to the spatial domain, participants showed reduced levels of uncertainty-directed exploration and increased levels of random exploration in the conceptual domain. Participants also displayed a one-directional transfer effect, where experience in the spatial task boosted performance in the conceptual task, but not vice versa. While confidence judgments indicated that participants were sensitive to the uncertainty of their knowledge in both tasks, they did not or could not leverage their estimates of uncertainty to guide exploration in the conceptual task. These results support the notion that value-guided learning and generalization recruit cognitive-map dependent computational mechanisms in spatial and conceptual domains. Yet both behavioral and model-based analyses suggest domain specific differences in how these representations map onto actions.

had no role in study design, data collection and analysis, decision to publish, or preparation of the manuscript.

**Competing interests:** The authors have declared that no competing interests exist.

## Author summary

There is a resurgence of interest in "cognitive maps" based on recent evidence that the hippocampal-entorhinal system encodes both spatial and non-spatial relational information, with far-reaching implications for human behavior. Yet little is known about the commonalities and differences in the computational principles underlying human learning and decision making in spatial and non-spatial domains. We use a within-subject design to examine how humans search for either spatially or conceptually correlated rewards. Using a Bayesian learning model, we find evidence for the same computational mechanisms of generalization across domains. While participants were sensitive to expected rewards and uncertainty in both tasks, how they leveraged this knowledge to guide exploration was different: participants displayed less uncertainty-directed and more random exploration in the conceptual domain. Moreover, experience with the spatial task improved conceptual performance, but not vice versa. These results provide important insights about the degree of overlap between spatial and conceptual cognition.

## Introduction

Thinking spatially is intuitive. We remember things in terms of places [1–3], describe the world using spatial metaphors [4, 5], and commonly use concepts like "space" or "distance" in mathematical descriptions of abstract phenomena. In line with these observations, previous theories have argued that reasoning about abstract conceptual information follows the same computational principles as spatial reasoning [6–8]. This has recently gained new support from neuroscientific evidence suggesting that common neural substrates are the basis for knowledge representation across domains [9–13].

One important implication of these accounts is that reinforcement learning [14] in non-spatial domains may rely on a map-like organization of information, supported by the computation of distances or similarities between experiences. These representations of distance facilitate generalization, allowing for predictions about novel stimuli based on their similarity to previous experiences. Here, we ask to what extent does the search for rewards depend on the same distance-dependent generalization across two different domains—one defined by spatial location and another by abstract features of a Gabor patch—despite potential differences in how the stimuli and their similarities may be processed?

We formalize a computational model that incorporates distance-dependent generalization and test it in a within-subject experiment, where either spatial features or abstract conceptual features are predictive of rewards. This allows us to study the extent to which the same organizational structure of cognitive representations is used in both domains, based on examining the downstream behavioral implications for learning, decision making, and exploration.

Whereas early psychological theories described reinforcement learning as merely developing an association between stimuli, responses, and rewards [15–17], more recent studies have recognized that the structure of representations plays an important role in making value-based decisions [11, 18] and is particularly important for knowing how to generalize from limited data to novel situations [19, 20]. This idea dates back to Tolman, who famously argued that both rats and humans extract a "cognitive map" of the environment [21]. This cognitive map encodes relationships between experiences or options, such as the distances between locations in space [22], and—crucially—facilitates flexible planning and generalization. While cognitive maps were first identified as representations of physical spaces, Tolman hypothesized that

similar principles may underlie the organization of knowledge in broader and more complex cognitive domains [21].

As was the case with Tolman, neuroscientific evidence for a cognitive map was initially found in the spatial domain, in particular, with the discovery of spatially selective place cells in the hippocampus [23, 24] and entorhinal grid cells that fire along a spatial hexagonal lattice [25]. Together with a variety of other specialized cell types that encode spatial orientation [26, 27], boundaries [28, 29], and distances to objects [30], this hippocampal-entorhinal machinery is often considered to provide a cognitive map facilitating navigation and self-location. Yet more recent evidence has shown that the same neural mechanisms are also active when reasoning about more abstract, conceptual relationships [31–36], characterized by arbitrary feature dimensions [37] or temporal relationships [38, 39]. For example, using a technique developed to detect spatial hexagonal grid-like codes in fMRI signals [40], Constantinescu et al. found that human participants displayed a pattern of activity in the entorhinal cortex consistent with mental travel through a 2D coordinate system defined by the length of a bird's legs and neck [9]. Similarly, the same entorhinal-hippocampal system has also been found to reflect the graph structure underlying sequences of stimuli [10] or the structure of social networks [41], and even to replay non-spatial representations in the sequential order that characterized a previous decision-making task [42]. At the same time, much evidence indicates that cognitive map-related representations are not limited to medial temporal areas, but also include ventral and orbital medial prefrontal areas [9, 11, 40, 43–45]. Relatedly, a study by Kahnt and Tobler [46] using uni-dimensional variations of Gabor stimuli showed that the generalization of rewards was modulated by dopaminergic activity in the hippocampus, indicating a role of non-spatial distance representations in reinforcement learning.

Based on these findings, we asked whether learning and searching for rewards in spatial and conceptual domains is governed by similar computational principles. Using a within-subject design comparing spatial and non-spatial reward learning, we tested whether participants used perceptual similarities in the same way as spatial distances to generalize from previous experiences and inform the exploration of novel options. To ensure commensurate stimuli discriminability between domains, participants completed a training phase where they were required to reach the same level of proficiency in correctly matching a series of target stimuli (see Methods; Fig 1c). In both domains, rewards were correlated (see S2 Fig), such that nearby or similar options tended to yield similar rewards. To model how participants generalize and explore using either perceptual similarities or spatial distances, we used Gaussian Process (GP) regression [47, 48] as a Bayesian model of generalization based on the principle of function learning. The Bayesian predictions of the GP model generalize about novel options using a common notion of similarity across domains, and provide estimates of expected reward and uncertainty. We tested out-of-sample predictions of the GP model against a Bayesian learner that incorporates uncertainty-guided exploration but without generalization, and investigated differences in parameters governing value-based decision making and uncertainty-directed exploration [49–51].

Participant performance was correlated across tasks and was best captured by the GP model in both domains. We were also able to reliably predict participant judgments about unobserved options based on parameters estimated from the bandit task. Whereas the model parameters indicated similar levels of generalization in both domains, we found lower levels of directed exploration in the conceptual domain, where participants instead showed increased levels of random exploration. Moreover, we also observed an asymmetric task order effect, where performing the spatial task first boosted performance on the conceptual task but not vice versa. These findings provide a clearer picture of both the commonalities and differences

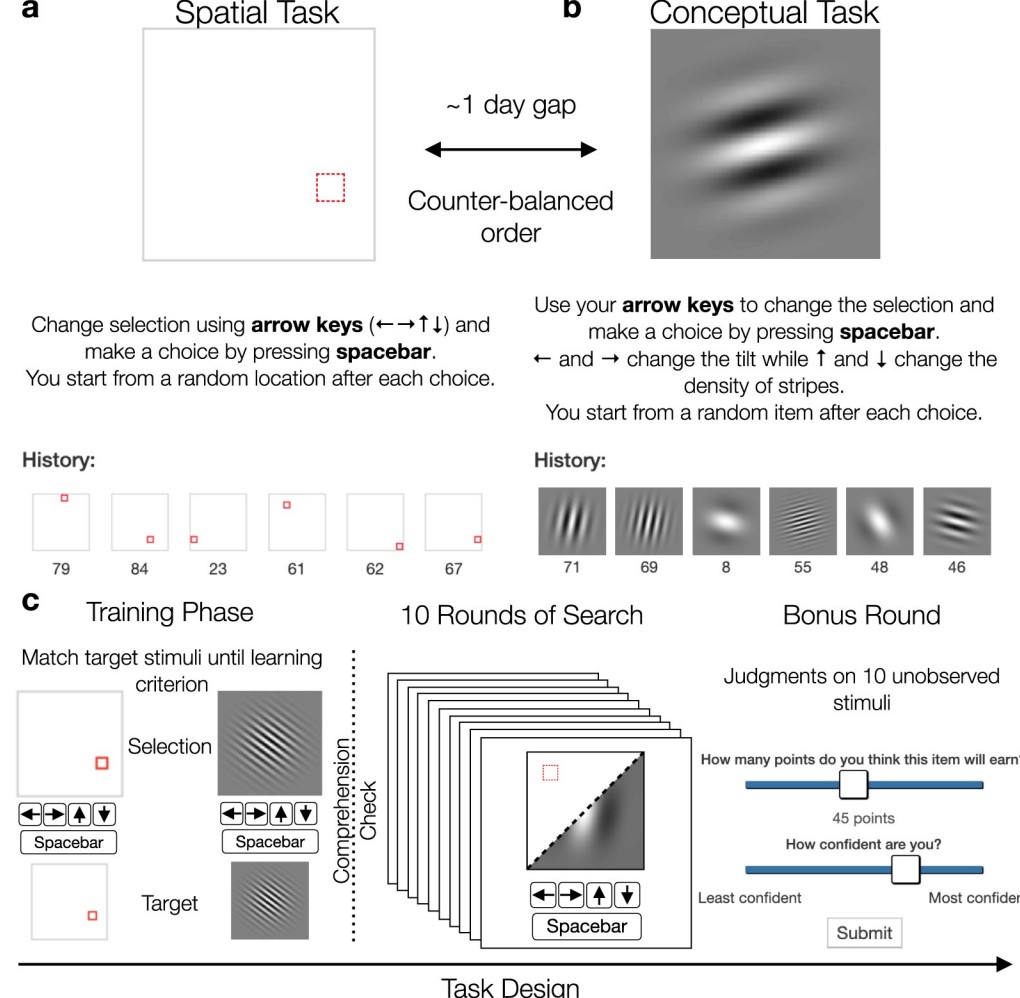

**Fig 1. Experiment design. a**) In the spatial task, options were defined as a highlighted square in a $8 \times 8$ grid, where the arrow keys were used to move the highlighted location. **b**) In the conceptual task, each option was represented as a Gabor patch, where the arrow keys changed the tilt and the number of stripes (S1 Fig). Both tasks corresponded to correlated reward distributions, where choices in similar locations or having similar Gabor features predicted similar rewards (S2 Fig). **c**) The same design was used in both tasks. Participants first completed a training phase where they were asked to match a series of target stimuli. This used the same inputs and stimuli as the main task, where the arrow keys modified either the spatial or conceptual features, and the spacebar was used to make a selection. After reaching the learning criterion of at least 32 training trials and a run of 9 out of 10 correct, participants were shown instructions for the main task and asked to complete a comprehension check. The main task was 10 rounds long, where participants were given 20 selections in each round to maximize their cumulative reward (shown in panels a and b). The 10th round was a "bonus round" where after 15 selections participants were asked to make 10 judgments about the expected reward and associated uncertainty for unobserved stimuli from that round. After judgments were made, participants selected one of the options, observed the reward, and continued the round as usual.

in how people reason about and represent both spatial and abstract phenomena in complex reinforcement learning tasks.

## Results

129 participants searched for rewards in two successive multi-armed bandit tasks (Fig 1). The *spatial* task was represented as an $8 \times 8$ grid, where participants used the arrow keys to move a highlighted square to one of the 64 locations, with each location representing one option (i.e.,

arm of the bandit). The *conceptual* task was represented using Gabor patches, where a single patch was displayed on the screen and the arrow keys changed the tilt and stripe frequency (each having 8 discrete values; see S1 Fig), providing a non-spatial domain where similarities are relatively well defined. Each of the 64 options in both tasks produced normally distributed rewards, where the means of each option were correlated, such that similar locations or Gabor patches with similar stripes and tilts yielded similar rewards (S2 Fig), thus providing traction for similarity-guided generalization and search. The strength of reward correlations were manipulated between subjects, with one half assigned to *smooth* environments (with higher reward correlations) and the other assigned to *rough* environments (with lower reward correlations). Importantly, both classes of environments had the same expectation of rewards across options.

The spatial and conceptual tasks were performed in counter-balanced order, with each task consisting of an initial training phase (see Methods) and then 10 rounds of bandits. Each round had a different reward distribution (drawn without replacement from the assigned class of environments), and participants were given 20 choices to acquire as many points as possible (later converted to monetary rewards). The search horizon was much smaller than the total number of options and therefore induced an explore-exploit dilemma and motivated the need for generalization and efficient exploration. The last round of each task was a "bonus round", where after 15 choices, participants were shown 10 unobserved options (selected at random) and asked to make judgments about the expected reward and their level of confidence (i.e., uncertainty about the expected rewards). These judgments were used to validate the internal belief representations of our models. All data and code, including interactive notebooks containing all analyses in the paper, is publicly available at https://github.com/charleywu/cognitivemaps.

## Computational models of learning, generalization, and search

Multi-armed bandit problems [52, 53] are a prominent framework for studying learning, where various reinforcement learning (RL) models [14] are used to model the learning of reward valuations and to predict behavior. A common element of most RL models is some form of prediction-error learning [54, 55], where model predictions are updated based on the difference between the predicted and experienced outcome. One classic example of learning from prediction errors is the Rescorla-Wagner [55] model, in which the expected reward $V(\cdot)$ of each bandit is described as a linear combination of weights $\mathbf{w}_t$ and a one-hot stimuli vector $\mathbf{x}_t$ representing the current state $s_t$:

$$V(\mathbf{x}_t) = \mathbf{w}_t^\top \mathbf{x}_t \tag{1}$$

$$\mathbf{w}_{t+1} = \mathbf{w}_t + \eta \delta_t \mathbf{x}_t \tag{2}$$

Learning occurs by updating the weights $\mathbf{w}$ as a function of the prediction error $\delta_t = r_t - V(\mathbf{x}_t)$, where $r_t$ is the observed reward, $V(\mathbf{x}_t)$ is the reward expectation, and $0 < \eta \leq 1$ is the learning rate parameter. In our task, we used a *Bayesian Mean Tracker* (BMT) as a Bayesian variant of the Rescorla-Wagner model [55, 56]. Rather than making point estimates of reward, the BMT makes independent and normally distributed predictions $V(s_{i,t}) \sim \mathcal{N}(m_{i,t}, v_{i,t})$ for each state $s_{i,t}$, which are characterized by a mean $m$ and variance $v$ and updated on each trial $t$ via the delta rule (see Methods for details).

**Generalization using Gaussian process regression.** Yet, an essential aspect of human cognition is the ability to generalize from limited experiences to novel options. Rather than learning independent reward representations for each state, we adopt a function learning

approach to generalization [19, 57], where continuous functions represent candidate hypotheses about the world, mapping the space of possible options to some outcome value. For example, a function can map how pressure on the gas pedal is related to the acceleration of a car, or how different amounts of water and fertilizer influence the growth rate of a plant. Crucially, the learned mapping provides estimates even for outcomes that have not been observed, by interpolating or extrapolating from previous experiences.

While the literature on how humans explicitly learn functions extends back to the 1960s [58], more recent approaches have proposed Gaussian Process (GP) regression [47] as a candidate model of human function learning [59–61]. GPs unite previous proposals of rule-based [62, i.e., learning the weights of a particular parametric function] and exemplar-based theories [63, i.e., neural networks predicting similar inputs will produce similar outputs], while also predicting the perceived difficulty of learning different functions [64] and explaining biases in how people extrapolate from limited data [59].

Formally, a GP defines a multivariate-normal distribution $P(f)$ over possible value functions $f(s)$ that map inputs $s$ to output $y = f(s)$.

$$P(f) \sim \mathcal{GP}(m(s), k(s, s')) \tag{3}$$

The GP is fully defined by the mean function $m(s)$, which is frequently set to 0 for convenience without loss of generality [47], and kernel function $k(s, s')$ encoding prior assumptions (or inductive biases) about the underlying function. Here we use the *radial basis function* (RBF) kernel:

$$k(s, s') = \exp\left(-\frac{\|s - s'\|^2}{2\lambda^2}\right) \tag{4}$$

encoding similarity as a smoothly decaying function of the squared Euclidean distance between stimuli $s$ and $s'$, measured either in spatial or conceptual distance. The length-scale parameter $\lambda$ encodes the rate of decay, where larger values correspond to broader generalization over larger distances.

Given a set of observations $\mathcal{D}_t = [\mathbf{s}_t, \mathbf{y}_t]$ about previously observed states and associated rewards, the GP makes normally distributed posterior predictions for any novel stimuli $s^\star$, defined in terms of a posterior mean and variance:

$$m(s^\star|\mathcal{D}_t) = K(s^\star, \mathbf{s}_t)[K(\mathbf{s}_t, \mathbf{s}_t) + \sigma_\epsilon^2 \mathbf{I}]^{-1} \mathbf{y}_t \tag{5}$$

$$v(s^\star|\mathcal{D}_t) = k(s^\star, s^\star) - K(s^\star, \mathbf{s}_t)[K(\mathbf{s}_t, \mathbf{s}_t) + \sigma_\epsilon^2 \mathbf{I}]^{-1} K(\mathbf{s}_t, s^\star) \tag{6}$$

The posterior mean corresponds to the expected value of $s^\star$ while the posterior variance captures the underlying uncertainty in the prediction. Note that the posterior mean can also be rewritten as a similarity-weighted sum:

$$m(s^\star|\mathcal{D}_t) = \sum_{i=1}^{t} w_i k(s^\star, s_i) \tag{7}$$

where each $s_i$ is a previously observed input in $\mathbf{s}_t$ and the weights are collected in the vector $\mathbf{w} = [K(\mathbf{s}_t, \mathbf{s}_t) + \sigma_\epsilon^2 \mathbf{I}]^{-1} \mathbf{y}_t$. Intuitively, this means that GP regression is equivalent to a linearly weighted sum, but uses basis functions $k(\cdot, \cdot)$ that project the inputs into a feature space, instead of the discrete state vectors. To generate new predictions, every observed reward $y_i$ in $\mathbf{y}_t$ is weighted by the similarity of the associated state $s_i$ to the candidate state $s^\star$ based on the kernel similarity. This similarity-weighted sum (Eq 7) is equivalent to a RBF network [65],

which has featured prominently in machine learning approaches to value function approximation [14] and as a theory of the neural architecture of human generalization [66] in vision and motor control.

**Uncertainty-directed exploration.**   In order to transform the Bayesian reward predictions of the BMT and GP models into predictions about participant choices, we use upper confidence bound (UCB) sampling together with a softmax choice rule as a combined model of both *directed* and *random* exploration [19, 50, 51].

UCB sampling uses a simple weighted sum of expected reward and uncertainty:

$$q_{UCB}(s) = m(s) + \beta\sqrt{v(s)} \qquad (8)$$

to compute a value $q$ for each option $s$, where the exploration bonus $\beta$ determines how to trade off exploring highly uncertain options against exploiting high expected rewards. This simple heuristic—although myopic—produces highly efficient learning by preferentially guiding exploration towards uncertain yet promising options, making it one of the only algorithms with known performance bounds in Bayesian optimization [67]. Recent studies have provided converging evidence for directed exploration in human behavior across a number of domains [19, 50, 68–70].

The UCB values are then put into a softmax choice rule:

$$P(s_i) = \frac{\exp\left(q(s_i)/\tau\right)}{\sum_j \exp\left(q(s_j)/\tau\right)} \qquad (9)$$

where the temperature parameter $\tau$ controls the amount of random exploration. Higher temperature sampling leads to more random choice predictions, with $\tau \rightarrow \infty$ converging on uniform sampling. Lower temperature values make more precise predictions, where $\tau \rightarrow 0$ converges on an argmax choice rule. Taken together, the exploration bonus $\beta$ and temperature $\tau$ parameters estimated on participant data allow us to assess the relative contributions of directed and undirected exploration, respectively.

## Behavioral results

After training participants were highly proficient in discriminating the stimuli, achieving at least 90% accuracy in both domains (see S3 Fig). Participants were also successful in both bandit tasks, achieving much higher rewards than chance in both conceptual (one-sample $t$-test: $t(128) = 24.6, p < .001, d = 2.2, BF > 100$) and spatial tasks ($t(128) = 34.6, p < .001, d = 3.0, BF > 100$; Fig 2a; See Methods for further details about statistics). In addition, participants could also leverage environmental structure in both domains. Using a two-way mixed ANOVA, we found that both environment (smooth vs. rough: $F(1, 127) = 9.4, p = .003, \eta^2 = .05, BF = 13$) and task (spatial vs. conceptual: $F(1, 127) = 35.8, p < .001, \eta^2 = .06, BF > 100$) influenced performance. The stronger reward correlations present in smooth environments facilitated higher performance (two sample $t$-test: $t(127) = 3.1, p = .003, d = 0.5, BF = 12$), even though both environments had the same expected reward.

While performance was strongly correlated between the spatial and conceptual tasks (Pearson's $r = .53, p < .001, BF > 100$; Fig 2b), participants performed systematically better in the spatial version (paired $t$-test: $t(128) = 6.0, p < .001, d = 0.5, BF > 100$). This difference in task performance can largely be explained by a one-directional transfer effect (Fig 2c). Participants performed better on the conceptual task after having experienced the spatial task ($t(127) = 2.8, p = .006, d = 0.5, BF = 6.4$). This was not the case for the spatial task, where performance did not differ whether performed first or second ($t(127) = -1.7, p = .096, d = 0.3, BF = .67$). Thus, experience with spatial search boosted performance on conceptual search, but not vice versa.

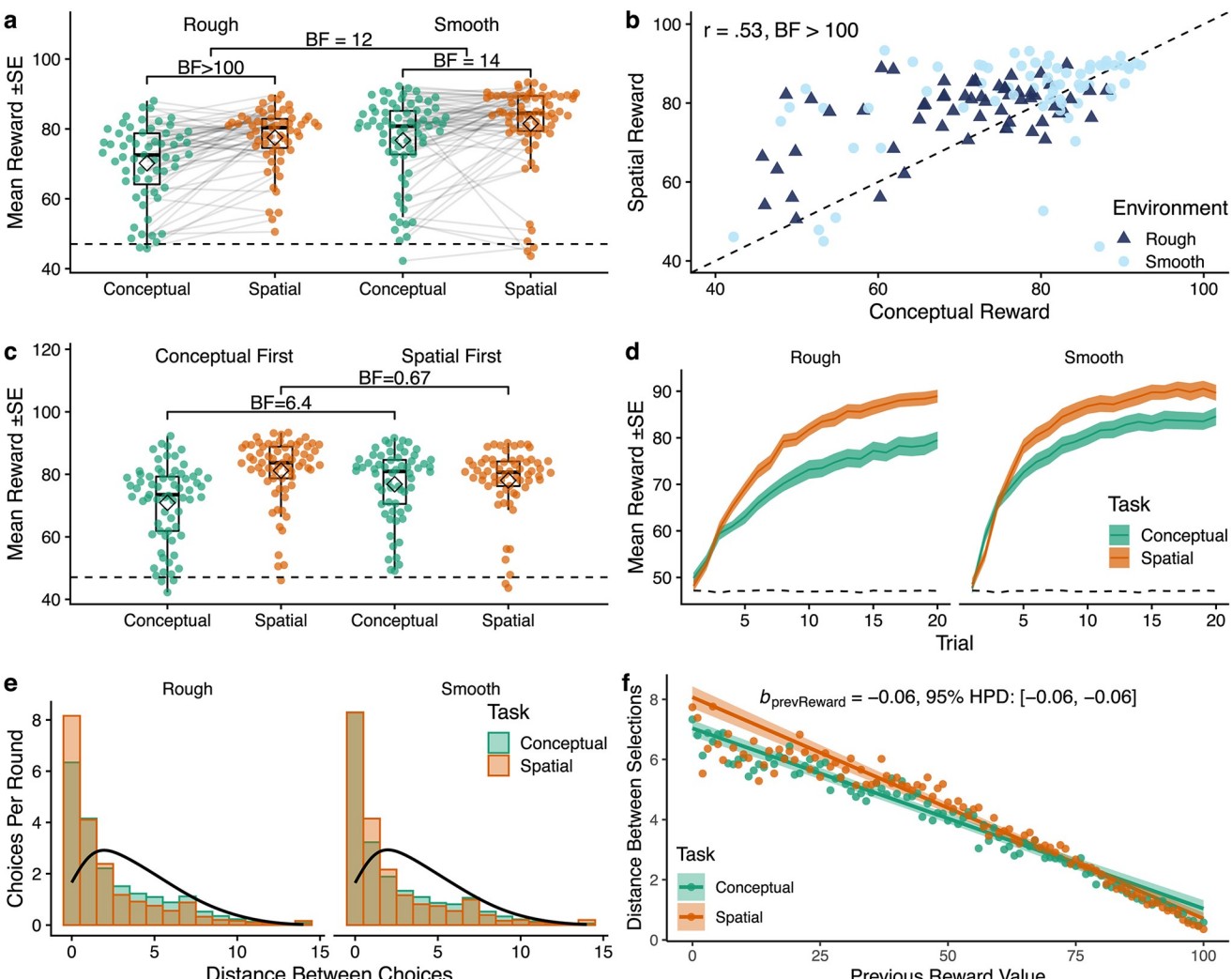

**Fig 2. Behavioral results. a)** Mean reward in each task, where each dot is a participant and lines connect the same participant across tasks. Tukey boxplots show median (horizontal line) and 1.5x IQR, while diamonds indicate the group mean. The dashed line indicates chance performance. Bayes Factors (*BF*) indicate the evidence against a specified null hypothesis for either two sample (rough vs. smooth) or paired (conceptual vs. spatial) *t*-tests (see Methods). **b)** Correspondence between tasks, where each dot represents the average reward of a single participant and the dotted line indicates *y* = *x*. **c)** Task order effect, where experience with spatial search boosted performance on conceptual search, but not vice versa. Bayes factors correspond to paired *t*-tests. **d)** Average learning curves over trials, showing the mean (line) and standard error (ribbon) aggregated across rounds and participants. The dashed line indicates chance performance. **e)** The Manhattan distance between selections compared to a random baseline (black line). **f)** Distance between selections as a function of the previous observed reward value, showing the aggregate means (points) and the group-level predictions of a mixed-effects regression (S1 Table), where the ribbons indicate the 95% CI.

Participants learned effectively within each round and obtained higher rewards with each successive choice (Pearson correlation between reward and trial: $r = .88$, $p < .001$, $BF > 100$; Fig 2d). We also found evidence for learning across rounds in the spatial task (Pearson correlation between reward and round: $r = .91$, $p < .001$, $BF = 15$), but not in the conceptual task ($r = .58$, $p = .104$, $BF = 1.5$).

Patterns of search also differed across domains. Comparing the average Manhattan distance between consecutive choices in a two-way mixed ANOVA showed an influence of task (within: $F(1, 127) = 13.8$, $p < .001$, $\eta^2 = .02$, $BF = 67$) but not environment (between: $F(1, 127) = 0.12$, $p = .73$, $\eta^2 = .001$, $BF = 0.25$, Fig 2e). This reflected that participants searched in smaller step

sizes in the spatial task ($t(128) = -3.7$, $p < .001$, $d = 0.3$, $BF = 59$), corresponding to a more local search strategy, but did not adapt their search distance to the environment. Note that each trial began with a randomly sampled initial stimuli, such that participants did not begin near the previous selection (see Methods). The bias towards local search (one-sample $t$-test comparing search distance against chance: $t(128) = -16.3$, $p < .001$, $d = 1.4$, $BF > 100$) is therefore not a side effect of the task characteristics, but both purposeful and effortful (see S4 Fig for additional analysis of search trajectories).

Participants also adapted their search patterns based on reward values (Fig 2f), where lower rewards predicted a larger search distance on the next trial (correlation between previous reward and search distance: $r = -.66$, $p < .001$, $BF > 100$). We analyzed this relationship using a Bayesian mixed-effects regression, where we found previous reward value to be a reliable predictor of search distance ($b_{prevReward} = -0.06$, 95% HPD: [−0.06, −0.06]; see S1 Table), while treating participants as random effects. This provides initial evidence for generalization-like behavior, where participants actively avoided areas with poor rewards and stayed near areas with rich rewards.

In summary, we found correlated performance across tasks, but also differences in both performance and patterns of search. Participants were boosted by a one-directional transfer effect, where experience with the spatial task improved performance on the conceptual task, but not the other way around. In addition, participants made larger jumps between choices in the conceptual task and searched more locally in the spatial task. However, participants adapted these patterns in both domains in response to reward values, where lower rewards predicted a larger jump to the next choice.

## Modeling results

To better understand how participants navigated the spatial and conceptual tasks, we used computational models to predict participant choices and judgments. Both GP and BMT models implement directed and undirected exploration using the UCB exploration bonus $\beta$ and softmax temperature $\tau$ as free parameters. The models differed in terms of learning, where the GP generalized about novel options using the length-scale parameter $\lambda$ to modulate the extent of generalization over spatial or conceptual distances, while the BMT learns the rewards of each option independently (see Methods).

Both models were estimated using leave-one-round-out cross validation, where we compared goodness of fit using out-of-sample prediction accuracy, described using a pseudo-$R^2$ (Fig 3a). The differences between models were reliable and meaningful, with the GP model making better predictions than the BMT in both the conceptual ($t(128) = 3.9$, $p < .001$, $d = 0.06$, $BF > 100$) and spatial tasks ($t(128) = 4.3$, $p < .001$, $d = 0.1$, $BF > 100$). In total, the GP model best predicted 85 participants in the conceptual task and 93 participants in the spatial task (out of 129 in total). Comparing this same out-of-sample prediction accuracy using a Bayesian model selection framework [71, 72] confirmed that the GP had the highest posterior probability (corrected for chance) of being the best model in both tasks (protected exceedance probability; conceptual: $pxp$(GP) = .997; spatial: $pxp$(GP) = 1.000; Fig 3b). The superiority of the GP model suggests that generalization about novel options via the use of structural information played a guiding role in how participants searched for rewards (see S6 Fig for additional analyses).

**Learning curves.**   To confirm that the GP model indeed captured learning behavior better in both tasks, we simulated learning curves from each model using participant parameter estimates (Fig 3c; see Methods). The GP model achieved human-like performance in all tasks and environments (comparing aggregate GP and human learning curves: conceptual MSE = 17.7;

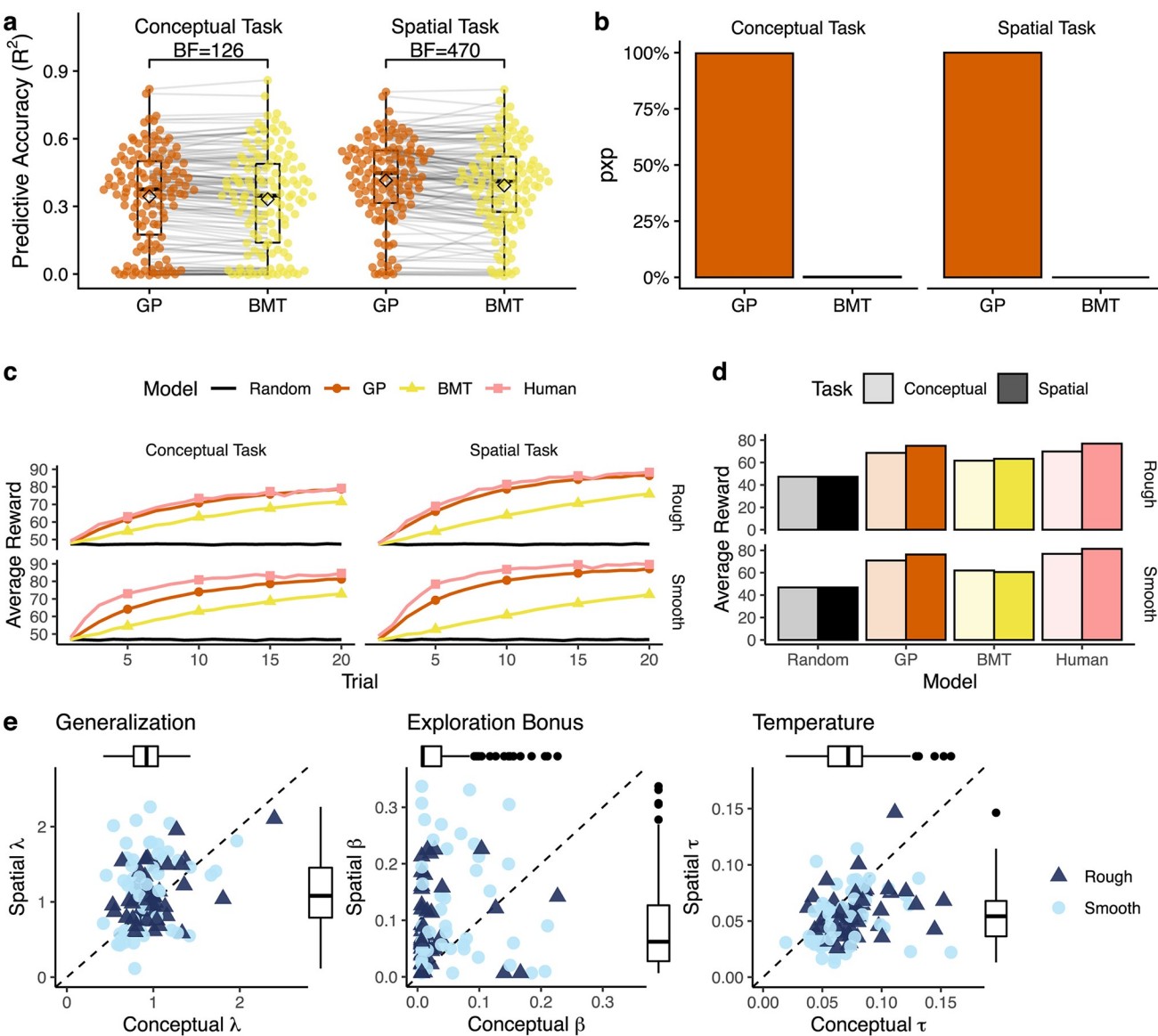

**Fig 3. Modeling results. a)** Predictive accuracy of each model, where 1 is a perfect model and 0 is equivalent to chance. Each dot is a single participant, with lines indicating the difference between models. Tukey boxplot shows the median (line) and 1.5 IQR, with the group mean indicated as a diamond. **b)** Protected Exceedence Probability (*pxp*), which provides a hierarchical estimate of model prevalence in the population (corrected for chance). **c)** Simulated learning curves. Each line is the averaged performance over 10,000 replications, where we sampled participant parameter estimates and simulated behavior on the task. The pink line is the group mean of our human participants, while the black line provides a random baseline. **d)** Simulation results from panel c aggregated over trials, where the height of the bar indicates average reward. **e)** GP parameter estimates from the conceptual (x-axis) and spatial (y-axis) tasks. Each point is the mean estimate for a single participant and the dotted line indicates *y = x*. For readability, the x- and y-axis limits are set to Tukey's upper fence (Q3 + 1.5 × IQR) for the larger of the two dimensions, but all statistics are performed on the full data.

spatial MSE = 16.6), whereas BMT learning curves were substantially less similar (conceptual MSE = 150.6; spatial MSE = 330.7). In addition, the GP captured the same qualitative difference between domains and environments as our human participants (Fig 3d), with better performance in conceptual vs. spatial, and smooth vs. rough. These patterns were not present in the BMT or random simulations.

**Parameter estimates.** To understand how generalization and exploration differed between domains, Fig 3e compares the estimated model parameters from the conceptual and

spatial tasks. The GP model had three free parameters: the extent of generalization ($\lambda$) of the RBF kernel, the exploration bonus ($\beta$) of UCB sampling, and the temperature ($\tau$) of the softmax choice rule (see S9 Fig for BMT parameters). Note that the exploration bonus captures exploration *directed* towards uncertainty, whereas temperature captures random, *undirected* exploration, which have been shown to be distinct and recoverable parameters [19, 70].

We do not find reliable differences in $\lambda$ estimates across tasks (Wilcoxon signed-rank test: $Z = -1.2$, $p = .115$, $r = -.11$, $BF = .13$). In all cases, we observed lower levels of generalization relative to the true generative model of the underlying reward distributions ($\lambda_{rough} = 2$, $\lambda_{smooth} = 4$; min-$BF = 1456$), replicating previous findings [19] that found undergeneralization to be largely beneficial in similar settings. Generalization was anecdotally correlated across tasks (Kendall rank correlation: $r_\tau = .13$, $p = .028$, $BF = 1.3$), providing weak evidence that participants tended to generalize similarly across domains.

Whereas generalization was similar between tasks, there were intriguing differences in exploration. We found substantially lower exploration bonuses ($\beta$) in the conceptual task ($Z = -5.0$, $p < .001$, $r = -.44$, $BF > 100$), indicating a large reduction of directed exploration, relative to the spatial task. At the same time, there was an increase in temperature ($\tau$) in the conceptual task ($Z = 6.9$, $p < .001$, $r = -.61$, $BF > 100$), corresponding to an increase in random, undirected exploration. These domain-specific differences in $\beta$ and $\tau$ were not influenced by task order or environment (two-way ANOVA: all $p > .05$, $BF < 1$). Despite these differences, we find some evidence of correlations across tasks for directed exploration ($r_\tau = .18$, $p = .002$, $BF = 13$) and substantial evidence for correlations between random exploration across domains ($r_\tau = .43$, $p < .001$, $BF > 100$).

Thus, participants displayed similar and somewhat correlated levels of generalization in both tasks, but with markedly different patterns of exploration. Whereas participants engaged in typical levels of directed exploration in the spatial domain (replicating previous studies; [19, 70]), they displayed reduced levels of directed exploration in the conceptual task, substituting instead an increase in undirected exploration. Again, this is not due to a lack of effort, because participants made longer search trajectories in the conceptual domain (see S4a Fig). Rather, this indicates a meaningful difference in how people represent or reason about spatial and conceptual domains in order to decide which are the most promising options to explore.

**Bonus round.** In order to further validate our behavioral and modeling results, we analyzed participants' judgments of expected rewards and perceived confidence for 10 unobserved options they were shown during the final "bonus" round of each task (see Methods and Fig 1c). Participants made equally accurate judgments in both tasks (comparing mean absolute error: $t(128) = -0.2$, $p = .827$, $d = 0.02$, $BF = .10$; Fig 4a), which were far better than chance (conceptual: $t(128) = -9.2$, $p < .001$, $d = 0.8$, $BF > 100$; spatial: $t(128) = -8.4$, $p < .001$, $d = 0.7$, $BF > 100$) and correlated between tasks ($r = .27$, $p = .002$, $BF = 20$). Judgment errors were also correlated with performance in the bandit task ($r = -.45$, $p < .001$, $BF > 100$), such that participants who earned higher rewards also made more accurate judgments.

Participants were equally confident in both domains ($t(128) = -0.8$, $p = .452$, $d = 0.04$, $BF = .13$; Fig 4b), with correlated confidence across tasks ($r = .79$, $p < .001$, $BF > 100$), suggesting some participants were consistently more confident than others. Ironically, more confident participants also had larger judgment errors ($r = .31$, $p < .001$, $BF = 91$) and performed worse in the bandit task ($r = -.28$, $p = .001$, $BF = 28$).

Using parameter estimates from the search task (excluding the entire bonus round), we computed model predictions for each of the bonus round judgments as an out-of-task prediction analysis. Whereas the BMT invariably made the same predictions for all unobserved options since it does not generalize (Fig 4c), the GP predictions were correlated with participant judgments in both conceptual (mean individual correlation: $\hat{r} = .35$; single sample $t$-test

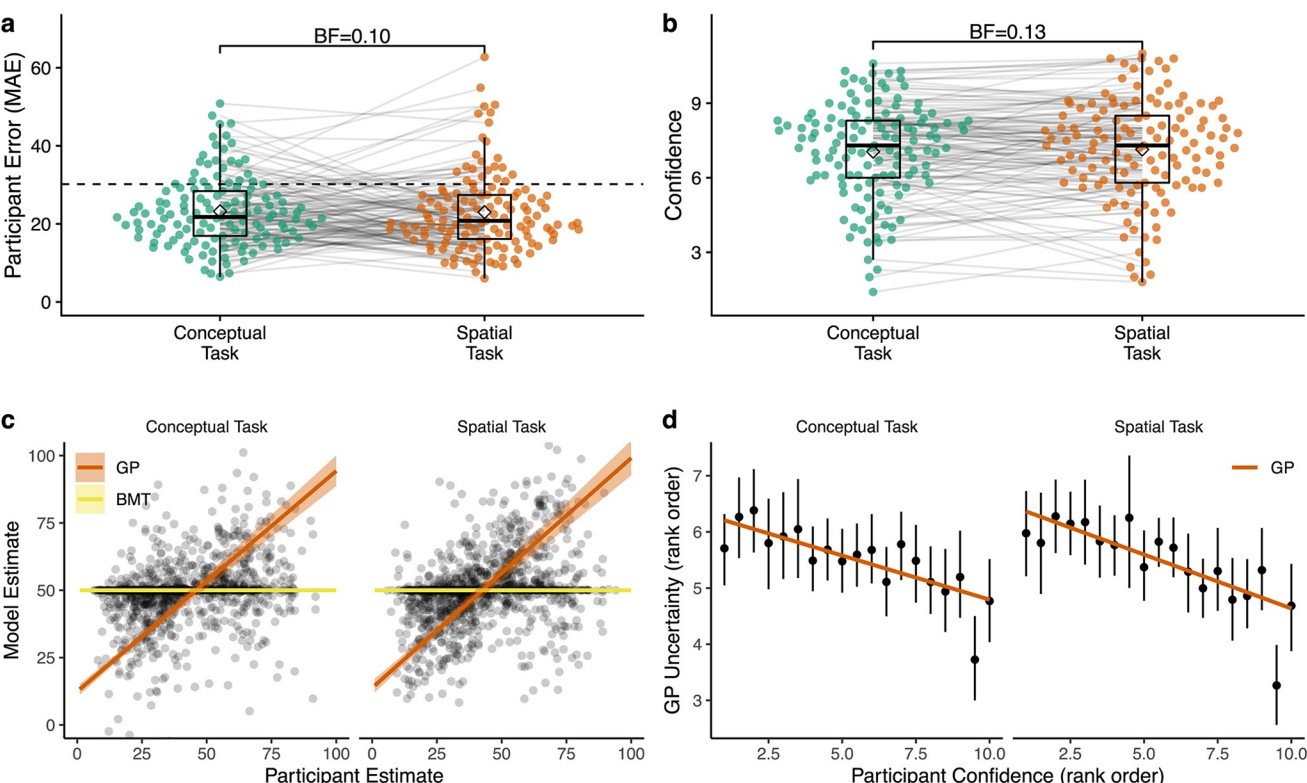

**Fig 4. Bonus round. a**) Mean absolute error (MAE) of judgments in the bonus round, where each dot is a single participant and lines connect performance across tasks. Tukey boxplot show median and $1.5 \times$ IQR, with the diamonds indicating group mean and the dashed line providing a comparison to chance. Bayes factor indicates the evidence against the null hypothesis for a paired $t$-test. **b**) Average confidence ratings (Likert scale: [0, 10]). **c**) Comparison between participant judgments and model predictions (based on the parameters estimated from the search task). Each point is a single participant judgment, with colored lines representing the predicted group-level effect of a mixed effect regression (S2 Table) and ribbons showing the 95% CI (undefined for the BMT model, which makes identical predictions for all unobserved options). **d**) Correspondence between participant confidence ratings and GP uncertainty, where both are rank-ordered at the individual level. Black dots show aggregate means and 95% CI, while the colored line is a linear regression.

of $z$-transformed correlation coefficients against $\mu = 0$: $t(128) = 11.0$, $p < .001$, $d = 1.0$, $BF > 100$) and spatial tasks ($\hat{r} = .43$; $t(128) = 11.0$, $p < .001$, $d = 1.0$, $BF > 100$). This correspondence between human judgments and model predictions was also confirmed using a Bayesian mixed effects model, where we again treated participants as random effects ($b_{\text{participantJudgment}} = .82$, 95% HPD: [0.75, 0.89]; see S2 Table for details).

Not only was the GP able to predict judgments about expected reward, but it also captured confidence ratings. Fig 4d shows how the highest confidence ratings corresponded to the lowest uncertainty estimates made by the GP model. This effect was also found in the raw data, where we again used a Bayesian mixed effects model to regress participant confidence judgments onto the GP uncertainty predictions ($b_{\text{participantJudgment}} = -0.02$, 95% HPD: [-0.02, -0.01]; see S2 Table).

Thus, participant search behavior was consistent with our GP model and we were also able to make accurate out-of-task predictions about both expected reward and confidence judgments using parameters estimated from the search task. These predictions validate the internal learning model of the GP, since reward predictions depend only on the generalization parameter $\lambda$. All together, our results suggest domain differences were not due to differences in how participants computed or represented expected reward and uncertainty, since they were equally good judging at their uncertainty in the bonus rounds for both domains. Rather, these diverging

patterns of search arose from differences in exploration, where participants substantially reduced their level of exploration directed towards uncertain options in the conceptual domain.

## Discussion

Previous theories of cognitive maps [21, 32–34] have argued that reasoning in abstract domains follows similar computational principles as in spatial domains, for instance, sharing a common approach to computing similarities between experiences. These accounts imply that the shared notion of similarity should influence how people generalize from past outcomes, and also how they balance between sampling new and informative options as opposed to exploiting known options with high expected rewards.

In addition, we investigated to what extent learning and searching for rewards are governed by similar computational principles in spatial and conceptual domains. Using a within-subject design, we studied participant behavior in both spatially and conceptually correlated reward environments. Comparing different computational models of learning and exploration, we found that a Gaussian Process (GP) model that incorporated distance-based generalization, and hence a cognitive map of similarities, best predicted participants behavior in both domains. In both domains, our parameter estimates indicated equivalent levels of generalization. Using these parameters, our model was able to simulate human-like learning curves and make accurate out-of-task predictions about participant reward estimations and confidence ratings in a final bonus round. This model-based evidence for similar distance-based decision making in both domains was also in line with our behavioral results. Performance was correlated across domains and benefited from higher outcome correlations between similar bandit options (i.e., smooth vs. rough). Subsequent choices tended to be more local than expected by chance, and similar options where more likely to be chosen after a high reward than a low reward outcome.

In addition to revealing similarities, our modelling and behavioral analyses provided a diagnostic lens into differences between spatial and conceptual domains. Whereas we found similar levels of generalization in both tasks, patterns of exploration were substantially different. Although participants showed clear signs of directed exploration (i.e., seeking out more uncertain options) in the spatial domain, this was notably reduced in the conceptual task. However, as if in compensation, participants increased their random exploration in the conceptual task. This implies a reliable shift in sampling strategies but not in generalization. Thus, even though the computational principles underpinning reasoning in both domains are indeed similar, how these computations are mapped onto actions can vary substantially. Moreover, participants obtained more rewards and sampled more locally in the spatial domain. We also find a one-directional transfer effect, where experience with the spatial task boosted performance on the conceptual task, but not vice versa. These findings shed new light onto the computational mechanisms of generalization and decision making, suggesting a universality of generalization and a situation-specific adaptation of decision making policies.

### Related work

Our findings also contribute to a number of other cognitive models and theories. According to the successor representation (SR; [18]) framework, hippocampal cognitive maps reflect predictions of expected future state occupancy [73–75]. This provides a similarity metric based on transition dynamics, where an analytic method for computing the SR in closed form is to assume random transitions through the state space. This assumption of a random policy produces a nearly identical similarity metric as the RBF kernel [76], with exact equivalencies in certain cases [77].

However, the SR can also be learned online using the Temporal-Difference learning algorithm, leading to asymmetric representations of distance that are skewed by the distance of travel [73, 78]. Recent work building on Kohonen maps has also suggested that the distribution of the experienced stimuli in feature space will have implications for the activation profiles of grid cells and the resulting cognitive map [79].

In our current study, we have focused on the simplifying case of a cognitive map learned through a random policy. This context was induced by having stimuli uniformly distributed over the search space and using a training phase involving extensive and random trajectories over the search space (i.e., matching random targets from random starting points). While this assumption is not always met in real life domains, it provides a useful starting point and allows us to reciprocally compare behavior in spatial and conceptual domains.

Previous work has also investigated transfer across domains [80], where inferences about the transition structure in one task can be generalized to other tasks. Whereas we used identical transition structures in both tasks, we nevertheless found asymmetric transfer between domains. A key question underlying the nature of transfer is the remapping of representations [81, 82], which can be framed as a hidden state-space inference problem. Different levels of prior experience with the spatial and conceptual stimuli could give rise to different preferences for reuse of task structure as opposed to learning a novel structure. This may be a potential source of the asymmetric transfer we measured in task performance.

Additionally, clustering methods (e.g., [79]) can also provide local approximations of GP inference by making predictions about novel options based on the mean of a local cluster. For instance, a related reward-learning task on graph structures [76] found that a $k$-nearest neighbors model provided a surprisingly effective heuristic for capturing aspects of human judgments and decisions. However, a crucial limitation of any clustering models is it would be incapable of learning and extrapolating upon any directional trends, which is a crucial feature of human function learning [59, 60]. Alternatively, clustering could also play a role in approximate GP inference [83], by breaking up the inference problem into smaller chunks or by considering only a subset of inputs. Future work should explore the question of how human inference scales with the complexity of the data.

Lastly, the question of "how the cognitive map is learned" is distinct from the question of "how the cognitive map is used". Here, we have focused on the latter, and used the RBF kernel to provide a map based on the assumption of random transitions, similar to a random-policy implementation of the SR. While both the SR and GP provide a theory of how people utilize a cognitive map for performing predictive inferences, only the GP provides a theory about representations of uncertainty via Bayesian predictions of reward. These representations of uncertainty are a key feature that sets the GP apart from the SR. Psychologically, GP uncertainty estimates systematically capture participant confidence judgments and provide the basis for uncertainty-directed exploration. This distinction may also be central to the different patterns of search we observed in spatial and non-spatial domains, where a reduction in uncertainty-directed exploration may also reflect computational differences in the structure of inference. However, the exact nature of these representations remains an open question for future neuro-imaging research.

## Future directions

Several questions about the link between cognitive maps across domains remain unanswered by our current study and are open for future investigations. Why did we find differences in exploration across domains, even though the tasks were designed to be as equivalent as possible, including requiring commensurate stimuli discriminability during the pre-task training

phase? Currently, our model can capture but not fully explain these differences in search behavior, since it treats both domains as equivalent generalization and exploration problems.

One possible explanation is a different representation of spatial and non-spatial information, or different computations acting on those representations. Recent experimental work has demonstrated that representations of spatial and non-spatial domains may be processed within the same neural systems [9, 12], suggesting representational similarities. But in our study it remains possible that different patterns of exploration could instead result from a different visual presentation of information in the spatial and the non-spatial task. It is, for example, conceivable that exploration in a (spatially or non-spatially) structured environment depends on the transparency of the structure in the stimulus material, or the alignment of the input modality. In our case the spatial structure was embedded in the stimulus itself, whereas the conceptual structure was not. Additionally, the arrow key inputs may have been more intuitive for manipulating the spatial stimuli. While generalization could be observed in both situations, directed exploration might require more explicitly accessible information about structural relationships or be facilitated by more intuitively mappable inputs. Previous work used a task where both spatial and conceptual features were simultaneously presented [84, i.e., conceptual stimuli were shuffled and arranged on a grid], yet only spatial or only conceptual features predicted rewards. However, differences in the saliency of spatial and conceptual features meant participants were highly influenced by spatial features, even when they were irrelevant. This present study was designed to overcome these issues by presenting only task-specific features, yet future work should address the computational features that allow humans to leverage structured knowledge of the environment to guide exploration. There is also a wide range of alternative non-spatial stimuli that we have not tested (for instance auditory [12] or linguistic stimuli [85, 86]), which could be considered more "conceptual" than our Gabor stimuli or may be more familiar to participants. Thus, it is an open empirical question to determine the limits to which spatial and different kinds of conceptual stimuli can be described using the same computational framework.

Our model also does not account for attentional mechanisms [87] or working memory constraints [88, 89], which may play a crucial role in influencing how people integrate information differently across domains [90]. To ask whether feature integration is different between domains, we implemented a variant of our GP model using a Shepard kernel [65], which used an additional free parameter estimating the level of integration between the two feature dimensions (S10 Fig). This model did not reveal strong differences in feature integration, yet replicated our main findings with respect to changes in exploration. Additional analyses showed asymmetries in attention to different feature dimensions, which was an effect modulated by task order (S4d–S4f Fig). Task order also modulated performance differences between domains, which only appeared when the conceptual task was performed before the spatial task (Fig 2c). Experience with the spatial task version may have facilitated a more homogenous mapping of the conceptual stimuli into a 2D similarity space, which in turn facilitated better performance. This asymmetric transfer may support the argument that spatial representations have been "exapted" to other more abstract domains [6–8]. For example, experience of different resource distributions in a spatial search task was found to influence behavior in a word generation task, where participants exposed to sparser rewards in space generated sparser semantic clusters of words [91]. Thus, while both spatial and conceptual knowledge are capable of being organized into a common map-like representation, there may be domain differences in terms of the ease of learning such a map and asymmetries in the transfer of knowledge. Future research should investigate this phenomenon with alternative models that make stronger assumptions about representational differences across domains.

We also found no differences in predictions and uncertainty estimates about unseen options in the bonus round. This means that participants generalized and managed to track the uncertainties of unobserved options similarly in both domains, yet did not or could not leverage their representations of uncertainty for performing directed exploration as effectively in the conceptual task. Alternatively, differences in random exploration could also arise from limited computational precision during the learning of action values [92]. Thus, the change in random exploration we observed may be due to different computational demands across domains. Similar shifts increases to random exploration have also been observed under direct cognitive load manipulations, such as by adding working memory load [93] or by limiting the available decision time [94].

Finally, our current experiment only looked at similarities between spatial and conceptual domains if the underlying structure was the same in both tasks. Future studies could expand this approach across different domains such as logical rule-learning, numerical comparisons, or semantic similarities. Additionally, structure learned in one domain could be transferable to structures encountered in either the same domain with slightly changed structures or even to totally different domains with different structures. A truly all-encompassing model of generalization should capture transfer across domains and structural changes. Even though several recent studies have advanced our understanding of how people transfer knowledge across graph structures [80], state similarities in multi-task reinforcement learning [95], and target hypotheses supporting generalization [90], whether or not all of these recruit the same computational principles and neural machinery remains to be seen.

## Conclusion

We used a rich experimental paradigm to study how people generalize and explore both spatially and conceptually correlated reward environments. While people employed similar principles of generalization in both domains, we found a substantial shift in exploration, from more uncertainty-directed exploration in the spatial task to more random exploration in the conceptual domain. These results enrich our understanding of the principles connecting generalization and search across different domains and pave the way for future cognitive and neuroscientific investigations.

## Methods

### Participants and design

140 participants were recruited through Amazon Mechanical Turk (requiring a 95% approval rate and 100 previously approved HITs) for a two part experiment, where only those who had completed part one were invited back for part two. In total 129 participants completed both parts and were included in the analyses (55 female; mean age = 35, SD = 9.5). Participants were paid $4.00 for each part of the experiment, with those completing both parts being paid an additional performance-contingent bonus of up to $10.00. Participants earned $15.6 ± 1.0 and spent 54 ± 19 minutes completing both parts. There was an average gap of 18 ± 8.5 hours between the two parts of the experiment.

We varied the task order between subjects, with participants completing the spatial and conceptual task in counterbalanced order in separate sessions. We also varied between subjects the extent of reward correlations in the search space by randomly assigning participants to one of two different classes of environments (*smooth* vs. *rough*), with smooth environments corresponding to stronger correlations, and the same environment class used for both tasks (see below).

### Ethics statement

The study was approved by the ethics committee of the Max Planck Institute for Human Development (A 2019/27) and all participants gave written informed consent.

### Materials and procedure

Each session consisted of a training phase, the main search task, and a bonus round. At the beginning of each session participants were required to complete a training task to familiarize themselves with the stimuli (spatial or conceptual), the inputs (arrow keys and spacebar), and the search space (8 × 8 feature space). Participants were shown a series of randomly selected targets and were instructed to use the arrow keys to modify a single selected stimuli (i.e., adjusting the stripe frequency and angle of a Gabor patch or moving the location of a spatial selector, Fig 1c) in order to match a target stimuli displayed below. The target stayed visible during the trial and did not have to be held in memory. The space bar was used to make a selection and feedback was provided for 800ms (correct or incorrect). Participants were required to complete at least 32 training trials and were allowed to proceed to the main task once they had achieved at least 90% accuracy on a run of 10 trials (i.e., 9 out of 10). See S3 Fig for analysis of the training data.

After completing the training, participants were shown instructions for the main search task and had to complete three comprehension questions (S11 and S12 Figs) to ensure full understanding of the task. Specifically, the questions were designed to ensure participants understood that the spatial or conceptual features predicted reward. Each search task comprised 10 rounds of 20 trials each, with a different reward function sampled without replacement from the set of assigned environments. The reward function specified how rewards mapped onto either the spatial or conceptual features, where participants were told that options with either similar spatial features (Spatial task) [19, 96] or similar conceptual features (Conceptual task) [20, 57] would yield similar rewards. Participants were instructed to accumulate as many points as possible, which were later converted into monetary payoffs.

The tenth round of each sessions was a "bonus round", with additional instructions shown at the beginning of the round. The round began as usual, but after 15 choices, participants were asked to make judgments about the expected rewards (input range: [1, 100]) and their level of confidence (Likert scale from least to most confident: [0, 10]) for 10 unrevealed targets. These targets were uniformly sampled from the set of unselected options during the current round. After the 10 judgments, participants were asked to make a forced choice between the 10 options. The reward for the selected option was displayed and the round continued as normal. All behavioral and computational modeling analyses exclude the last round, except for the analysis of the bonus round judgments.

**Spatial and conceptual search tasks.**   Participants used the arrow keys to either move a highlighted selector in the spatial task or change the features (tilt and stripe frequency) of the Gabor stimuli in the conceptual task (S1 Fig). On each round, participants were given 20 trials to acquire as many cumulative rewards as possible. A selection was made by pressing the space bar, and then participants were given feedback about the reward for 800 ms, with the chosen option and reward value added to the history at the bottom of the screen. At the beginning of each trial, the starting position of the spatial selector or the displayed conceptual stimulus was randomly sampled from a uniform distribution. Each reward observation included normally distributed noise, $\epsilon \sim \mathcal{N}(0, 1)$, where the rewards for each round were scaled to a uniformly sampled maximum value in the range of 80 to 95, so that the value of the global optima in each round could not be easily guessed.

Participants were given feedback about their performance at the end of each round in terms of the ratio of their average reward to the global maximum, expressed as a percentage (e.g., "You have earned 80% of the maximum reward you could have earned on this round"). The performance bonus (up to $10.00) was calculated based on the cumulative performance of each round and across both tasks.

**Bonus round judgments.**   In both tasks the last round was a "bonus round", which solicited judgments about the expected reward and their level of confidence for 10 unrevealed options. Participants were informed that the goal of the task remained the same (maximize cumulative rewards), but that after 15 selections, they would be asked to provide judgments about 10 randomly selected options, which had not yet been explored. Judgments about expected rewards were elicited using a slider from 1 to 100 (in increments of 1), while judgments about confidence were elicited using a slider from 0 to 10 (in increments of 1), with the endpoints labeled 'Least confident' and 'Most confident'. After providing the 10 judgments, participants were asked to select one of the options they just rated, and subsequently completed the round like all others.

**Environments.**   All environments were sampled from a GP prior parameterized with a *radial basis function* (RBF) kernel (Eq 4), where the length-scale parameter ($\lambda$) determines the rate at which the correlations of rewards decay over (spatial or conceptual) distance. Higher $\lambda$-values correspond to stronger correlations. We generated 40 samples of each type of environments, using $\lambda_{rough} = 2$ and $\lambda_{smooth} = 4$, which were sampled without replacement and used as the underlying reward function in each task (S2 Fig). Environment type was manipulated between subjects, with the same environment type used in both conceptual and spatial tasks.

## Models

**Bayesian mean tracker.**   The Bayesian Mean Tracker (BMT) is a simple but widely-applied associative learning model [69, 97, 98], which is a special case of the Kalman Filter with time-invariant reward distributions. The BMT can also be interpreted as a Bayesian variant of the Rescorla-Wagner model [56], making predictions about the rewards of each option $j$ in the form of a normally distributed posterior:

$$P(\mu_{j,t}|\mathcal{D}_t) = \mathcal{N}(m_{j,t}, v_{j,t}) \tag{10}$$

The posterior mean $m_{j,t}$ and variance $v_{j,t}$ are updated iteratively using a delta-rule update based on the observed reward $y_t$ when option $j$ is selected at trial $t$:

$$m_{j,t} = m_{j,t-1} + \delta_{j,t}G_{j,t}[y_t - m_{j,t-1}] \tag{11}$$

$$v_{j,t} = [1 - \delta_{j,t}G_{j,t}]v_{j,t-1} \tag{12}$$

where $\delta_{j,t} = 1$ if option $j$ was chosen on trial $t$, and 0 otherwise. Rather than having a fixed learning rate, the BMT scales updates based on the Kalman Gain $G_{j,t}$, which is defined as:

$$G_{j,t} = \frac{v_{j,t-1}}{v_{j,t-1} + \theta_\epsilon^2} \tag{13}$$

where $\theta_\epsilon^2$ is the error variance, which is estimated as a free parameter. Intuitively, the estimated mean of the chosen option $m_{j,t}$ is updated based on the prediction error $y_t - m_{j,t-1}$ and scaled by the Kalman Gain $G_{j,t}$ (Eq 11). At the same time, the estimated variance $v_{j,t}$ is reduced by a factor of $1 - G_{j,t}$, which is in the range [0, 1] (Eq 12). The error variance $\theta_\epsilon^2$ can be interpreted as

an inverse sensitivity, where smaller values result in more substantial updates to the mean $m_{j,t}$, and larger reductions of uncertainty $v_{j,t}$.

## Model cross-validation

As with the behavioral analyses, we omit the 10th "bonus round" in our model cross-validation. For each of the other nine rounds, we use cross validation to iteratively hold out a single round as a test set, and compute the maximum likelihood estimate using differential evolution [99] on the remaining eight rounds. Model comparisons use the summed out-of-sample prediction error on the test set, defined in terms of log loss (i.e., negative log likelihood).

**Predictive accuracy.** As an intuitive statistic for goodness of fit, we report *predictive accuracy* as a pseudo-$R^2$:

$$R^2 = 1 - \frac{\log \mathcal{L}(M_k)}{\log \mathcal{L}(M_{rand})} \tag{14}$$

comparing the out-of-sample log loss of a given model $M_k$ against a random model $M_{rand}$. $R^2 = 0$ indicates chance performance, while $R^2 = 1$ is a theoretically perfect model.

**Protected exceedance probability.** The protected exceedance probability (*pxp*) is defined in terms of a Bayesian model selection framework for group studies [71, 72]. Intuitively, it can be described as a random-effect analysis, where models are treated as random effects and are allowed to differ between subjects. Inspired by a Polya's urn model, we can imagine a population containing $K$ different types of models (i.e., people best described by each model), much like an urn containing different colored marbles. If we assume that there is a fixed but unknown distribution of models in the population, what is the probability of each model being more frequent in the population than all other models in consideration?

This is modelled hierarchically, using variational Bayes to estimate the parameters of a Dirichlet distribution describing the posterior probabilities of each model $P(m_k|\mathbf{y})$ given the data $\mathbf{y}$. The exceedance probability is thus defined as the posterior probability that the frequency of a model $r_{m_k}$ is larger than all other models $r_{m_{k' \neq k}}$ under consideration:

$$xp(m_k) = p(r_{m_k} > r_{m_{k' \neq k}}|\mathbf{y}) \tag{15}$$

[72] extends this approach by correcting for chance, based on the Bayesian Omnibus Risk (*BOR*), which is the posterior probability that all model frequencies are equal:

$$pxp(m_k) = xp(m_k)(1 - BOR) + \frac{BOR}{K} \tag{16}$$

This produces the *protected exceedance probability* (*pxp*) reported throughout this article, and is implemented using https://github.com/sjgershm/mfit/blob/master/bms.m.

**Simulated learning curves.** We simulated each model by sampling (with replacement) from the set of cross-validated participant parameter estimates, and performing search on a simulated bandit task. We performed 10,000 simulations for each combination of model, environment, and domain (spatial vs. conceptual).

**Bonus round predictions.** Bonus round predictions used each participant's estimated parameters to predict their judgments about expected reward and confidence. Because rewards in each round were randomly scaled to a different global maximum, we also rescaled the model predictions in order to align model predictions with the observed rewards and participant judgments.

## Statistical tests

**Comparisons.** We report both frequentist and Bayesian statistics. Frequentist tests are reported as Student's *t*-tests (specified as either paired or independent) for parametric comparisons, while the Mann-Whitney-*U* test or Wilcoxon signed-rank test are used for non-parametric comparisons (for independent samples or paired samples, respectively). Each of these tests are accompanied by a Bayes factors (*BF*) to quantify the relative evidence the data provide in favor of the alternative hypothesis ($H_A$) over the null ($H_0$), which we interpret following [100].

Parametric comparison are tested using the default two-sided Bayesian *t*-test for either independent or dependent samples, where both use a Jeffreys-Zellner-Siow prior with its scale set to $\sqrt{2}/2$, as suggested by [101]. All statistical tests are non-directional as defined by a symmetric prior (unless otherwise indicated).

Non-parametric comparisons are tested using either the frequentist Mann-Whitney-*U* test for *independent samples*, or the Wilcoxon signed-rank test for *paired samples*. In both cases, the Bayesian test is based on performing posterior inference over the test statistics (Kendall's $r_\tau$ for the Mann-Whitney-*U* test and standardized effect size $r = \frac{z}{\sqrt{N}}$ for the Wilcoxon signed-rank test) and assigning a prior using parametric yoking [102]. This leads to a posterior distribution for Kendall's $r_\tau$ or the standardized effect size $r$, which yields an interpretable Bayes factor via the Savage-Dickey density ratio test. The null hypothesis posits that parameters do not differ between the two groups, while the alternative hypothesis posits an effect and assigns an effect size using a Cauchy distribution with the scale parameter set to $1/\sqrt{2}$.

**Correlations.** For testing linear correlations with Pearson's *r*, the Bayesian test is based on Jeffrey's [103] test for linear correlation and assumes a shifted, scaled beta prior distribution $B\left(\frac{1}{k}, \frac{1}{k}\right)$ for *r*, where the scale parameter is set to $k = \frac{1}{3}$ [104].

For testing rank correlations with Kendall's tau, the Bayesian test is based on parametric yoking to define a prior over the test statistic [105], and performing Bayesian inference to arrive at a posterior distribution for $r_\tau$. The Savage-Dickey density ratio test is used to produce an interpretable Bayes Factor.

**ANOVA.** We use a two-way mixed-design analysis of variance (ANOVA) to compare the means of both a fixed effects factor (smooth vs. rough environments) as a between-subjects variable and a random effects factor (conceptual vs. spatial) as a within-subjects variable. To compute the Bayes Factor, we assume independent g-priors [106] for each effect size $\theta_1 \sim \mathcal{N}(0, g_1\sigma^2), \cdots, \theta_p \sim \mathcal{N}(0, g_p\sigma^2)$, where each g-value is drawn from an inverse chi-square prior with a single degree of freedom $g_i \overset{i.i.d}{\sim} \text{inverse} - \chi^2(1)$, and assuming a Jeffreys prior on the aggregate mean and scale factor. Following [107], we compute the Bayes factor by integrating the likelihoods with respect to the prior on parameters, where Monte Carlo sampling was used to approximate the g-priors. The Bayes factor reported in the text can be interpreted as the log-odds of the model relative to an intercept-only null model.

**Mixed effects regression.** Mixed effects regressions are performed in a Bayesian framework with `brms` [108] using MCMC methods (No-U-Turn sampling [109] with the proposal acceptance probability set to.99). In all models, we use a maximal random effects structure [110], and treat participants as a random intercept. Following [111] we use the following generic weakly informative priors:

$$b_0 \sim \mathcal{N}(0, 1) \tag{17}$$

$$b_i \sim \mathcal{N}(0, 1) \tag{18}$$

$$\sigma \sim \text{Half-}\mathcal{N}(0, 1) \tag{19}$$

All models were estimated over four chains of 4000 iterations, with a burn-in period of 1000 samples.

## Supporting information

**S1 Fig. Gabor stimuli.** Tilt varies from left to right from 105˚ to 255˚ in equally spaced intervals, while stripe frequency increases moving upwards from 1.5 to 15 in log intervals. (TIFF)

**S2 Fig. Correlated reward environments.** Heatmaps of the reward environments used in both spatial and conceptual domains. The color of each tile represents the expected reward of the bandit, where the x-axis and y-axis were mapped to the spatial location or the tilt and stripe frequency (respectively). All environments have the same minimum and maximum reward values, and the two classes of environments share the same expectation of reward across options. (EPS)

**S3 Fig. Training phase. a**) Trials needed to reach the learning criterion (90% accuracy over 10 trials) in the training phase, where the dotted line indicates the 32 trial minimum. Each dot is a single participant with lines connecting the same participant. Tukey boxplots show median (line) and 1.5x IQR, with diamonds indicating group means. **b**) Average correct choices during the training phase. In the last 10 trials before completing the training phase, participants had a mean accuracy of 95.0% on the spatial task and 92.7% on the conceptual task (difference of 2.3%). In contrast, in the first 10 trials of training, participants had a mean accuracy of 84.1% in the spatial task and 68.8% in the conceptual (difference of 15.4%). **c**) Heatmaps of the accuracy of different target stimuli, where the x and y-axes of the conceptual heatmap indicate tilt and stripe frequency, respectively. **d**) The probability of error as a function of the magnitude of error (Manhattan distance from the correct response). Thus, most errors were close to the target, with higher magnitude errors being monotonically less likely to occur. (EPS)

**S4 Fig. Search trajectories. a**) Distribution of trajectory length, separated by task and environment. The dashed vertical line indicates the median for each category. Participants had longer trajectories in the contextual task ($t(128) = -10.7$, $p < .001$, $d = 1.0$, $BF > 100$), but there were no differences across environments ($t(127) = 1.3$, $p = .213$, $d = 0.2$, $BF = .38$). **b**) Average reward value as a function of trajectory length. Longer trajectories were correlated with higher rewards ($r = .23$, $p < .001$, $BF > 100$). Each dot is a mean with error bars showing the 95% CI. **c**) Distance from the random initial starting point in each trial as a function of the previous reward value. Each dot is the aggregate mean, while the lines show the fixed effects of a Bayesian mixed-effects model (see S1 Table), with the ribbons indicating the 95% CI. The relationship is not quite linear, but is also found using a rank correlation ($r_\tau = .18$, $p < .001$, $BF > 100$). The dashed line indicates random chance. **d**) Search trajectories decomposed into the vertical/stripe frequency dimension vs. horizontal/tilt dimension. Bars indicate group means and error bars show the 95% CI. We find more attention given to the vertical/stripe frequency dimension in both tasks, with a larger effect for the conceptual task ($F(1, 127) = 26.85$, $p < .001$, $\eta^2 = .08$, $BF > 100$), but no difference across environments ($F(1, 127) = 1.03$, $p = .311$, $\eta^2 = .005$, $BF = 0.25$). **e**) We compute attentional bias as $\Delta_{\text{dim}} = P(\text{vertical/stripe frequency}) - P(\text{horizontal/tilt})$, where positive values indicate a stronger bias towards the vertical/stripe frequency

dimension. Attentional bias was influenced by the interaction of task order and task ($F(1, 127)$ = 8.1, $p$ = .005, $\eta^2$ = .02, $BF > 100$): participants were more biased towards the vertical/stripe frequency dimension in the conceptual task when the conceptual task was performed first ($t$ (66) = − 6.0, $p < .001$, $d = 0.7$, $BF > 100$), but these differences disappeared when the spatial task was performed first ($t(61)$ = − 1.6, $p$ = .118, $d = 0.2$, $BF$ = .45). **f**) Differences in attention and score. Each participant is represented as a pair of dots, where the connecting line shows the change in score and $\Delta_{dim}$ across tasks. We found a negative correlation between score and attention for the conceptual task only in the conceptual first order ($r_\tau$ = − .31, $p < .001$, $BF > 100$), but not in the spatial first order ($r_\tau$ = − .07, $p$ = .392, $BF$ = .24). There were no relationships between score and attention in the spatial task in either order (spatial first: $r_\tau$ = .03, $p$ = .738, $BF$ = .17; conceptual first: $r_\tau$ = − .03, $p$ = .750, $BF$ = .17).
(EPS)

**S5 Fig. Heatmaps of choice frequency.** Heatmaps of chosen options in **a**) the Gabor feature of the conceptual task and **b**) the spatial location of the spatial task, aggregated over all participants. The color shows the frequency of each option centered on yellow representing random chance (1/64), with orange and red indicating higher than chance, while green and blue were lower than chance.
(EPS)

**S6 Fig. Additional modeling results. a**) The relationship between mean performance and predictive accuracy, where in all cases, the best performing participants were also the best described. **b**) The best performing participants were also the most diagnostic between models, but not substantially skewed towards either model. Linear regression lines strongly overlap with the dotted line at $y = 0$, where participants above the line were better described by the GP model. **c** Model comparison split by which task was performed first vs. second. In both cases, participants were better described on their second task, although the superiority of the GP over the BMT remains, comparing only task one (paired $t$-test: $t(128)$ = 4.6, $p < .001$, $d = 0.10$, $BF$ = 1685) or only task two ($t(128)$ = 3.5, $p < .001$, $d = 0.08$, $BF$ = 27).
(EPS)

**S7 Fig. GP parameters and performance. a**) We do not find a consistent relationship between $\lambda$ estimates and performance, which were anecdotally correlated in the spatial task ($r_\tau$ = .13, $p$ = .030, $BF$ = 1.2) or negatively correlated in the conceptual task ($r_\tau$ = − .22, $p < .001$, $BF > 100$). **b**) Higher $\beta$ estimates were strongly predictive of better performance in both conceptual ($r_\tau$ = .32, $p < .001$, $BF > 100$) and spatial tasks ($r_\tau$ = .31, $p < .001$, $BF > 100$). **c**) On the other hand, high temperature values predicted lower performance in both conceptual($r_\tau$ = − .59, $p < .001$, $BF > 100$) and spatial tasks ($r_\tau$ = − .58, $p < .001$, $BF > 100$).
(EPS)

**S8 Fig. GP exploration bonus and temperature.** We check here whether there exists any inverse relationship between directed and undirected exploration, implemented using the UCB exploration bonus $\beta$ (x-axis) and the softmax temperature $\tau$ (y-axis), respectively. Results are split into conceptual (**a**) and spatial tasks (**b**), where each dot is a single participant and the dotted line indicates $y = x$. The upper axis limits are set to the largest $1.5 \times$ IQR, for both $\beta$ and $\tau$, across both conceptual and spatial tasks.
(EPS)

**S9 Fig. BMT parameters.** Each dot is a single participant and the dotted line indicates $y = x$. **a**) We found lower error variance ($\sigma_\epsilon^2$) estimates in the conceptual task (Wilcoxon signed-rank test: $Z$ = − 4.8, $p < .001$, $r$ = − .42, $BF > 100$), suggesting participants were more sensitive to

the reward values (i.e., more substantial updates to their means estimates). Error variance was also correlated across tasks ($r_\tau$ = .18, $p$ = .003, $BF$ = 10). **b**) As with the GP model reported in the main text, we also found strong differences in exploration behavior in the BMT. We found lower estimates of the exploration bonus in the conceptual task ($Z$ = − 5.9, $p$ < .001, $r$ = − .52, $BF$ > 100). The exploration bonus was also somewhat correlated between tasks ($r_\tau$ = .16, $p$ = .006, $BF$ = 4.8). **c**) Also in line with the GP results, we again find an increase in random exploration in the conceptual task ($Z$ = − 6.9, $p$ < .001, $r$ = − .61, $BF$ > 100). Once more, temperature estimates were strongly correlated ($r_\tau$ = .34, $p$ < .001, $BF$ > 100).
(EPS)

**S10 Fig. Shepard kernel parameters.** We also considered an alternative form of the GP model. Instead of modeling generalization as a function of squared-Euclidean distance with the RBF kernel, we use the Shepard kernel described in [65], where we instead use Minkowski distance with the free parameter $\rho \in [0, 2]$. This model is identical to the GP model reported in the main text when $\rho = 2$. But when $\rho < 2$, the input dimensions transition from integral to separable representations [112]. The lack of clear differences in model parameters motivated us to only include the standard RBF kernel in the main text. **a**) We find no evidence for differences in generalization between tasks ($Z$ = − 1.8, $p$ = .039, $r$ = − .15, $BF$ = .32). There is also marginal evidence of correlated estimates ($r_\tau$ = .13, $p$ = .026, $BF$ = 1.3). **b**) There is anecdotal evidence of lower $\rho$ estimates in the conceptual task ($Z$ = − 2.5, $p$ = .006, $r$ = − .22, $BF$ = 2.0). The implication of a lower $\rho$ in the conceptual domain is that the Gabor features were treated more independently, whereas the spatial dimensions were more integrated. However, the statistics suggest this is not a very robust effect. These estimates are also not correlated ($r_\tau$ = − .02, $p$ = .684, $BF$ = .12). **c**) Consistent with all the other models, we find systematically lower exploration bonuses in the conceptual task ($Z$ = − 5.5, $p$ < .001, $r$ = − .49, $BF$ > 100). There was weak evidence of a correlation across tasks ($r_\tau$ = .14, $p$ = .021, $BF$ = 1.6). **d**) We find clear evidence of higher temperatures in the conceptual task ($Z$ = − 6.3, $p$ < .001, $r$ = − .56, $BF$ > 100), with strong correlations across tasks ($r_\tau$ = .41, $p$ < .001, $BF$ > 100).
(EPS)

**S11 Fig. Comprehension questions for the conceptual task.** The correct answers are highlighted.
(TIFF)

**S12 Fig. Comprehension questions for the spatial task.** The correct answers are highlighted.
(TIFF)

**S1 Table. Mixed effects regression results: Previous reward.**
(PDF)

**S2 Table. Mixed effects regression results: Bonus round judgments.**
(PDF)

## Acknowledgments

We thank Daniel Reznik, Nicholas Franklin, Samuel Gershman, Christian Doeller, and Fiery Cushman for helpful discussions.

## Author Contributions

**Conceptualization:** Charley M. Wu, Eric Schulz, Mona M. Garvert, Björn Meder, Nicolas W. Schuck.

**Formal analysis:** Charley M. Wu.

**Software:** Charley M. Wu.

**Visualization:** Charley M. Wu.

**Writing – original draft:** Charley M. Wu, Eric Schulz, Mona M. Garvert, Björn Meder, Nicolas W. Schuck.

**Writing – review & editing:** Charley M. Wu, Eric Schulz, Mona M. Garvert, Björn Meder, Nicolas W. Schuck.

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
