## [Decision Letter · Decision Letter 0]

5 Mar 2020

Dear Mr. Wu,

Thank you very much for submitting your manuscript "Similarities and differences in spatial and non-spatial cognitive maps" for consideration at PLOS Computational Biology.

As with all papers reviewed by the journal, your manuscript was reviewed by members of the editorial board and by several independent reviewers.

The paper was overall well received, but some important issues need to be addressed, in particular involving better motivating your approach and situating it in the state of the art, together with commenting on its generalizability.

Concerning the statistics and the report of the results:

- please always report all the data points, as you do in most figures, instead of bar plots with confidence bars

- removing outliers messes with the degrees of freedom. Several alternative approaches exist, excellent robust alternatives are proposed in this paper Wilcox, R. R., & Rousselet, G. A. (2018). A Guide to Robust Statistical Methods in Neuroscience. Current Protocols in Neuroscience, 82(1). doi:10.1002/cpns.41 (open access here https://www.biorxiv.org/content/10.1101/151811v1). The same paper also suggests multivariate robust linear regression as an alternative to ANOVA

- some data appear to be distributed in a very non-linear way, questioning the linear fit

In light of the reviews (below this email), we would like to invite the resubmission of a significantly-revised version that takes into account the reviewers' comments.

We cannot make any decision about publication until we have seen the revised manuscript and your response to the reviewers' comments. Your revised manuscript is also likely to be sent to reviewers for further evaluation.

Sincerely,

Daniele Marinazzo

Deputy Editor

PLOS Computational Biology

Daniele Marinazzo

Deputy Editor

PLOS Computational Biology

Reviewer's Responses to Questions

**Comments to the Authors:**

Reviewer #1: The manuscript “Similarities and differences in spatial and non-spatial cognitive maps” claims that spatial and conceptual cognitive maps are fundamentally different. The authors propose that the two domains are dealt with differently in important and characteristic ways. One of the main differences is claimed to be that conceptual cognition involves more random exploration while spatial cognition involves more uncertainty-driven exploration. The authors also discovered that experience with space helps with concepts (higher accuracy), but not vice versa. To explain and analyse their results the authors use a Gaussian process model.

Overall, I am unsure about how generalisable these results are, given the authors have not sufficiently proposed any strong theoretical constraints on their hypotheses. I would have loved to have seen a higher-level theoretical account as to why they designed their experiment the way they did, why they suspected sampling strategies would differ (or perhaps they did not?), and especially why they chose the stimuli used. I will elaborate below what I mean and how I think some of these issues can be addressed.

Firstly, in terms of the stimuli: I believe that Gabor patches are not confirmed to be homogeneously mappable to 2D. This might indicate that while participants can achieve high accuracy in the training task (which was matching a stimulus to target stimulus by moving in 2D space) the higher cognitive demands of the test phase (which involves exploration as well) would impair participants’ accuracy if Gabor patches are harder to map onto 2D. In other words, due to the nature of the stimuli the test phase could be harder for Gabor patches. To assuage my worries, the authors could show that the stimuli in the two cases (spatial vs conceptual) are indeed 2D and indeed homogeneously distributed in their respective domains. What I mean is that Gabor patches might not all be equally easy to tell apart as a function of their frequency and orientation. And therefore, arguably, the spatial case could be seen as more homogenous. One way to explore this is to first ascertain if the stimuli are 2D — something like Ahlheim and Love (2018, the code is open source) could be run on the raw pixels of what the participants see to ensure both sets of stimuli have the same dimensionality. Alternatively, other methods of investigating this are possible. After that, assuming that both spaces are found to be (roughly) 2D, the issue of the homogeneity of the spaces can be addressed (thank you to Sebastian Bobadilla Suarez for input on this issue). Is moving e.g., one step in 2D frequency/orientation space also one step in the stimulus space of the Gabor stimuli? Mutatis mutandis for the spatial case, of course, which I suspect is 2D and homogenous.

Secondly, as mentioned above, I believe that some attention needs to be paid to other models, like Kohonen maps, e.g., the work in Mok and Love (2019). It might prove useful to give a few sentences on such models’ computational properties in order to understand what the paper sets out to investigate: the computational overlap between special and conceptual cognitive processing and what such an overlap might imply. In other words, given the authors are interested in the computational nature of cognitive maps some mention of modelling maps (explicitly) computationally is pertinent. Furthermore, it might help address, or at least contextualise, some of the ideas around the one-directional facilitation effect found and provide a formalisable structure and plan for future work.

Thirdly, if all the above is addressed, I would be more comfortable with the claims that this is a “fundamental difference in how people represent or reason about spatial and conceptual domains” but still not completely. Arguably people have vastly more experience with a 2D spatial domain than the domain of Gabor patches. Even the input modality is more easily mappable onto the spatial than conceptual domain since participants used the keyboard arrows in both tasks. I do not believe this is a fatal flaw in the paper, but it is something that has to be touched on: input space and task space are aligned more so in the spatial case than the concept case.

Code: I am having trouble running your code. I suggest the first step is to tidy up your code according to R best practices and especially in terms of dependencies by including a DESCRIPTION file and name these requirements, explaining how to install them in your README file. Also mention the version of R you used to create and run your codebase. See: http://r-pkgs.had.co.nz/description.html#dependencies — as well as: https://github.com/ropensci/Rclean and https://github.com/ironholds/urltools as examples of good practice to copy from.

Minor: Figure 3 panel d has a typo, should be “Bonus”.

References

Ahlheim, C., & Love, B. C. (2018). Estimating the functional dimensionality of neural representations. NeuroImage, 179, 51-62.

Mok, R. M., & Love, B. C. (2019). A non-spatial account of place and grid cells based on clustering models of concept learning. Nature communications, 10(1), 1-9.

Reviewer #2: I enjoyed reading this paper. I think trying to understand computational differences in how individuals reason and generalize in spatial and non-spatial maps explore is an important problem in cognitive science. The study presented in the paper finds some intriguing similarities and differences between generalization and exploration in these domains that I think will be inspiring for future research. In general, the analysis are very well presented and I appreciate the thorough investigation of the behavioral data in a model-free manner in addition to the sophisticated model-fitting. I only have a few critiques.

1. Comparisons in generalization and exploration parameters between the two tasks rely on distances between the two stimuli meaning the same thing between the two tasks. There’s no reason though that this should inherently be the case. The authors use evidence from the training task to argue that there are not perceptual discriminability differences between the stimuli for the two tasks. I’m not sure I understood though how this would address the question of whether distances between stimuli are comparable.

As a side point here, I’m not sure I fully understood the training task and what exactly data from it is meant to show. Was the target on the screen while the subjects navigated to it, or were subjects required to hold the target in memory? Additionally, in order to receive a correct response, were subjects required to take the shortest path to the target, or merely to arrive at the target eventually - perhaps this would bear on whether subjects intuited a map-like distance in the conceptual space that is similar to the map distance in spatial space?

Related to the question about how we can know whether distances between tasks are equivalent, I’m not sure it’s fair to assume that distances across the two dimensions of the conceptual stimuli mean the same thing (as i think is assumed by the RBF kernel in the GP model.) I think the authors should either address this with further analysis, or discuss whether this assumption being untrue would change interpretation of parameters from the model.

If it is not possible to address these concerns with further analysis, I think the paper is still valuable and interesting, but I think the authors should address, in the discussion, whether this concern poses limitations in the interpretation of parameter similarities and differences.

2. I appreciated the in depth model-free analysis (Behavioral Results section) prior to the modeling analysis. However it seems that a number of the features of the data that are presented in the model-free are not addressed again in the modeling section. This leads to the impression that perhaps there are aspects of behavior that the GP model is not picking up.

In particular, I was wondering whether differences in model parameters (between environments and also between tasks) can account for the following features in the data:

- That participants get more rewards in smooth compared to rough domains

- The one-directional transfer effect that subjects conceptual performance benefits from first performing the spatial task.

Relatedly, in comparing human and model learning curves (figure 3c) it appears that humans outperform the model in smooth, but not in rough environments. Why does the model fail to capture learning curves as well in smooth environments?

I think it is fine if the model cannot account for all these differences. But it would be useful for the reader for the paper to clearly state what aspects of the model-free analysis the GP models can and cannot account for.

3. Lastly, for comparison of exploration parameters, I think the difference in directed exploration between environments is a really interesting difference between tasks. However, I question whether it is fair to interpret differences in the random exploration parameter as a strategy difference. This is because, I presume, that errors in model prediction of behavior get soaked up into that parameter. If this is correct, couldn’t the paper equivalently just state that the model fits worse in the conceptual task than the spatial task?

Reviewer #3: Wu and colleagues tested human participants on a spatial and conceptual task to assess whether similar cognitive mechanisms are used across these domains. They applied computational modelling and found shared and different processes, suggesting some processes are shared whereas other processes might be distinct.

This is an exceptionally well conducted study with a clear rationale, strong analyses and modelling work. I believe this will be a great paper for PLOS computational biology, after minor revisions. Mainly, I have some questions to clarify parts of the paper, and on how the works compares to some of the current literature. I also include some few suggestions that I hope will help the paper, if space allows.

Task design

1. Smooth vs rough designs: Is the reason why people get more rewards on smooth conditions because there is more rewards overall across the map, or is it actually that they do better in smooth environments because they learnt it?

I ask because it looks like smooth environments have more rewards (looking at the maps in S2 - more yellow cells). But it also sounds like it is normalized so overall same expected reward - so would that mean each yellow cell means lower rewards in the smooth versus rough? But if that's the case then it's much harder to get higher reward in the smooth conditions (since highest reward per choice is lower?)

Results

2. Do participants attend to one feature dimension more than another (e.g. Nosofsky, 1986 - you could check if they weighted one dimension more than another / more sensitive to one dimension)? Probably not for spatial, but maybe in the gabors? Does that affect anything / maybe harder to generalize to space if so?

3. Related to the above point - transfer - is there an analysis to show why there is transfer for space  gabors but not vice versa? E.g. general: people who learn better on space  transfer more. If so - maybe people who learn better on gabors also transfer more to space (even though main effect not there). Possibly, those who show more 'equal' attention to both dimensions on gabors show better transfer?

Model results:

4. The results of 3A are convincing but shows that the BMT model does well as well; whereas 3B looks like it's doing very poorly; why is this? I realize A is predicting novel choices, and B is model fit. Is BMT actually doing a very bad job of fitting, but still getting pretty good predictive accuracy?

Model questions / considerations:

5. Do the rewards have to be spatially correlated for the GP model to work well / generalize? Comparing the BMT (point estimate) versus the GP model - it makes sense that the function learning approach will do better than a point estimate approach when locations in the space are correlated. Would your approach do as well if they were less/not (spatially) correlated? i.e. would the GP model still work with learning structures that were not, e.g. distributed smoothly?

My question is - if there is structure, but not spatially smooth at all (rough is still quite smooth), would the GP still learn it? Eg. Across blocks, you change the rewarded locations, but the relations between the reward locations are kept the same. Or would a point-estimate model do as well?

6. In the Discussion: "Comparing different computational models of learning and exploration, we found that a Gaussian Process model that incorporated distance-based generalization, and hence a cognitive map of similarities, best predicted participants behavior in both domains. "

The authors only compare with one model - BMT, a Bayesian model that does point estimates on rewards. How about other models? The claim that GP is a good model because it captures the generalization bit is fine - but maybe less emphasis on the model as the only model? E.g. can other models solve it? Maybe there are good reasons this model is better theoretically anyway - could discuss this and why it's better/ different to other models at least

Theoretically, would a gaussian mixture model or clustering model not also work for structures where the rewards are spatially correlated like this? You'd get generalization to new parts of the space if they learn the centres of the reward regions - though I'm not sure if it would learn as quickly as is needed (compared to GPs)

To be clear, I am not asking the authors to run all the models, but state what they show (they can show good generalization, but not that this is the only model can do that). It could make sense to discuss other models and maybe why they won't work if that is the case.

Related literature

7. How is it related to other ideas about the neural underpinnings, e.g. grid cells? For example, the successor representation (e.g. Stachenfeld et al., 2017, Mommenejad & Howard, 2018), clustering (Mok & Love, 2019), Gaussian/Bayesian mixture models and more (e.g. Sanders, Wilson, Gershman 2019, bioRxiv), spatial-conceptual (Bellmund et al., 2019- cited but relevant comparison). Would these models do well at your task, or is GP the only one that could capture the data? Are there any predictions or interpretations of the GP for neural data?

Suggestions

Abstract:

1. Key findings include both similarities and differences between cognitive mechanisms for the two tasks - the differences are described well but the similarities are a bit vague: "Using a Bayesian learning model, we find evidence for the same computational mechanisms of generalization across domains." If there is space, I suggest the authors could add a sentence or state what they find - that there are no/little differences between the parameters from these model across tasks, and they are correlated across participants (or qualify which parameters were not different/correlated).

Introduction

2. An explanation of what 'generalization' in cited work and in the current work is assumed. Something simple would already help: e.g. when people learn or gain an understanding about an environment, they can generalize in the sense they know what the value of the novel options are and select them even though they have never experienced them.

3. How is cognitive maps/generalization related to directed vs random exploration? Not sure this is addressed in the introduction, though the authors set out to test it.

4. The reader would benefit from a short introduction to GPs and the motivation for using them - why are they a good model to be used here? How are they related to previous ideas mentioned in the intro? Could be very brief and since there is more in the Results.

Minor:

5. Suggestion: Figure 1 - why not show all the gabors to illustrate the structure of the stimulus space, to show the correspondence to the spatial structure? It is not clear that this is the case. Could be nice to have the supplement figure S1 included here, if it fits.

**Have all data underlying the figures and results presented in the manuscript been provided?**

Reviewer #1: Yes

Reviewer #2: Yes

Reviewer #3: Yes

PLOS authors have the option to publish the peer review history of their article (what does this mean?). If published, this will include your full peer review and any attached files.

Reviewer #1: Yes: Olivia Guest

Reviewer #2: No

Reviewer #3: No
---

## [Decision Letter · Decision Letter 1]

25 May 2020

Dear Mr. Wu,

Thank you very much for submitting your manuscript "Similarities and differences in spatial and non-spatial cognitive maps" for consideration at PLOS Computational Biology.

We appreciate the changes that you made to your manuscript following the recommendations, and two reviewers are fully satisfied by them.

On the other hand Dr. Guest feels that some major concerns have not been addressed, and I agree. Now, it can be that there is a misunderstanding in the conception of these issues, or in the wording used to address them, or even that you don't feel like that these issues need to be addressed, or to be addressed as suggested by Dr. Guest. Or a mixture of all this.

I think that it's important to disambiguate these issues and to agree (even agree to disagree) on the remaining issues.

We cannot make any decision about publication until we have seen the revised manuscript and your response to the reviewers' comments. Your revised manuscript is also likely to be sent to reviewers for further evaluation.

Sincerely,

Daniele Marinazzo

Deputy Editor

PLOS Computational Biology

Reviewer's Responses to Questions

**Comments to the Authors:**

Reviewer #1: My review is uploaded as a PDF attachment.

Reviewer #2: The authors have adequately addressed all of my concerns. I think the revised paper is strong.

Reviewer #3: Thank you for writing this detailed and clear response, engaging with my questions, and the effort on all the extra analyses.

The authors have answered all my questions and have improved the paper with clarifications and additional analyses.

**Have all data underlying the figures and results presented in the manuscript been provided?**

Reviewer #1: Yes

Reviewer #2: Yes

Reviewer #3: Yes

PLOS authors have the option to publish the peer review history of their article (what does this mean?). If published, this will include your full peer review and any attached files.

Reviewer #1: Yes: Olivia Guest

Reviewer #2: No

Reviewer #3: No
---

## [Decision Letter · Decision Letter 2]

13 Jul 2020

Dear Mr. Wu,

We are pleased to inform you that your manuscript 'Similarities and differences in spatial and non-spatial cognitive maps' has been provisionally accepted for publication in PLOS Computational Biology.

Some disagreements remain, and since in general reviewers are proxies of the wider community of readers, I hope you will agree to make the reviews public, and to engage with any subsequent comment (this should be true in general for any research product anyway).

Best regards,

Daniele Marinazzo

Deputy Editor

PLOS Computational Biology

Daniele Marinazzo

Deputy Editor

PLOS Computational Biology

Reviewer's Responses to Questions

**Comments to the Authors:**

Reviewer #1: I thank the authors for apologising for their confusing previous response letter, however I am not entirely sure this more recent one is a huge improvement. I am not used to dealing with differences between phraseology in the letter versus the manuscript. Ideally, the authors should be consistent because both manuscript and letter reflect their internal ideas about their work. Are they going to publish a very carefully written piece (this one) but then talk about it in presentations using language that is so generalised that they will get the reactions they got from me in my previous reply? (Anyway, this is a rhetorical question for the editor and authors.)

Some of the changes to the manuscript are appropriate and useful — I am glad the authors spent time attempting to improve their paper — while in other cases the authors have opted not to modify or clarify their prose. Given this, they likely do not want to edit their manuscript (in those cases) because they disagree with my perspectives as given in my previous review, which is of course understandable and totally within their remit. So while I think some of the changes made are appropriate, some of the more important points I raised still stand.

To end, I appreciate some of the changes the authors have made, but we obviously disagree on some, perhaps very core, issues. Since it’s not a productive use of time to just rehash our previous disagreements (which are in my previous review, of course), I will close by saying that I wish the authors and their manuscript all the best and that I have nothing further to add.

**Have all data underlying the figures and results presented in the manuscript been provided?**

Reviewer #1: Yes

PLOS authors have the option to publish the peer review history of their article (what does this mean?). If published, this will include your full peer review and any attached files.

Reviewer #1: **Yes: **Olivia Guest

---

## [Editor Report · Acceptance letter]

20 Aug 2020

PCOMPBIOL-D-20-00183R2 

Similarities and differences in spatial and non-spatial cognitive maps

Dear Dr Wu,

I am pleased to inform you that your manuscript has been formally accepted for publication in PLOS Computational Biology. Your manuscript is now with our production department and you will be notified of the publication date in due course.

With kind regards,

Matt Lyles
